# Model Simulation and Rheological Research on Crosslinking Behavior of Polyethylene Resin

**DOI:** 10.3390/gels10010035

**Published:** 2023-12-31

**Authors:** Xuelian Chen, Qigu Huang

**Affiliations:** 1State Key Laboratory of Chemical Resource Engineering, MOE Key Laboratory of Carbon Fibers and Functional Polymers, The College of Material Science and Engineering, Beijing University of Chemical Technology, Beijing 100029, China; 2Shenhua (Beijing) New Materials Technology Co., Ltd., CHN Energy Group, Beijing 102211, China

**Keywords:** polyethylene, crosslink, polymeric structure, model simulation, rheological method

## Abstract

The crosslinking behavior of polyethylene (PE) determines its exceptional performance and application. In this study, we investigated the crosslinking behaviors of different PE resins through model simulation and rheological methods. Specifically, the mathematical equation of “S” model was established for PE resin. According to this equation, the optimal maximum gel content for high-density polyethylene (HDPE) was found to be around 85%. Moreover, the maximum crosslinking degrees for different PE resins depended largely on their density and molecular weight. The melt viscosities before crosslinking in PE resins were highly influenced by their melt index. The higher melt indexes resulted in the lower storage moduli, improving melt processability during processing. In addition, the crosslinking rates of PE resins were strongly influenced by peroxide concentration, independent of PE resin structures. For high molecular weight and low-density PE resins, they exhibited decreased ti values, increased A_0_ values, and decreased k6 values. However, there were no noticeable variations in the values of k2 and phi among different PE resins. All simulated modeling outcomes showed remarkable consistency with the experimental rheological data. These findings are of strong significance in the industrial manufacture of PE resin.

## 1. Introduction

Polyethylene (PE) is a widely used material and one of the most popular materials in engineering. It is a thermoplastic polymer that is valued for its versatility and durability. PE is commonly used in various applications such as packaging, construction, transportation, and electrical engineering [1,2,3]. PE is known for its high strength-to-weight ratio, making it an ideal choice for manufacturing lightweight but sturdy products. It is also resistant to moisture, chemicals, and impact, further enhancing its suitability for a wide range of applications. In addition to its physical properties, PE is also a highly versatile material that can be easily molded into different shapes and sizes. This makes it a cost-effective option for manufacturers and engineers, as it can be customized to meet specific design requirements. Furthermore, PE is likely to remain an essential material in various industries for years to come. Its continued popularity and demand highlight its significance in the field of engineering.

Crosslinking is a crucial process in the creation of PE materials. It involves the formation of covalent bonds between polymer chains, which results in the development of a three-dimensional macromolecular structure. This modification of the polymer structure is typically achieved through various chemical methods [4,5,6,7,8]. When PE chains are linked through crosslinking, it enhances the physical properties of the material, such as its strength, durability, and thermal stability. This makes crosslinked polymers highly desirable for a wide range of applications, including in the automotive, aerospace, and construction industries [9,10,11,12,13]. Crosslinked polyethylene (XLPE) has become increasingly popular among scientists and engineers for its outstanding performance properties. XLPE exhibits excellent cable function, high wear resistance, exceptional chemical resistance, superior notched impact strength, minimal shrinkage, and superior thermal resistance. These properties make XLPE a highly desirable material for a wide range of applications. One of the key advantages of XLPE is its excellent cable function. XLPE cables are widely used in power distribution and transmission due to their low dielectric loss, high insulation resistance, and excellent electrical properties. These cables also exhibit high mechanical strength and are resistant to moisture and environmental stress, making them suitable for outdoor and underground applications. In addition to its exceptional cable function, XLPE also offers high wear resistance, making it an ideal material for applications requiring durability and longevity. Its resistance to wear and abrasion allows it to withstand harsh operating conditions, making it a preferred material for pipes, fittings, and other components in industrial and infrastructure applications [14,15,16]. Additionally, XLPE can enhance the rheological properties and foamability of PE [17,18,19]. Generally, there are three main methods for crosslinking: high-energy radiation crosslinking [20], organic peroxide-induced chemical crosslinking [21], and silane–water crosslinking [22]. Among these, the chemical crosslinking reaction using organic peroxide is widely favored for its cost-effectiveness and minimal by-products. However, the challenge in using chemical crosslinkable PE to produce high-performance plastic product lies in controlling and preventing premature crosslinking. Research involving molecular dynamics simulations and experimental techniques has indicated that a high ratio of peroxide to PE can increase the production of byproducts, but does not necessarily result in a higher amount of gel fraction. To date, the understanding of chemical crosslinking has been limited due to the complexity of its chemical reaction [23,24]. Various reaction mechanisms have been suggested for rubbers and the most widely accepted one is as follows [25,26,27,28]. The peroxide complex initially decomposes to generate active radicals at high temperatures. Subsequently, the radicals are transferred to the hydrogen atom site in the PE molecule, based on electronegativity differences. Finally, the PE molecules with the radicals react with each other and form crosslinking bonds. However, there are often side reactions during the crosslinking process. It remains a significant challenge to understand the effects of PE types and peroxide complex content on pre-crosslinking melt processability, crosslinking rate, and the final degree of crosslinking.

For the research and development of PE resin, it is essential to conduct thorough kinetic analysis and use rheological methods. These processes provide valuable insights into the behavior of the resin under various conditions, helping to optimize its properties and enhance its performance. Kinetic analysis involves studying the reaction rates and mechanisms involved in the production of PE resin. This includes determining the optimal conditions for polymerization, identifying key intermediates and by-products, and understanding the overall kinetics of the process. By gaining a deep understanding of the reaction kinetics, researchers can fine-tune the production process to achieve the desired molecular structure and properties of the resin [29,30]. Rheological methods, on the other hand, focus on the study of the flow and deformation behavior of PE resin. This includes measuring parameters such as viscosity, elasticity, and viscoelasticity under different temperatures, pressures, and shear rates. By analyzing the rheological properties of the resin, researchers can gain valuable insights into its processability, melt stability, and overall performance in various applications [31]. Furthermore, research on kinetic analysis and rheological methods for PE resin also involves the investigation of additives and fillers that can modify its properties. This includes studying the effects of various additives on the reaction kinetics and rheological behavior of the resin, as well as understanding their impact on the final properties of the material. In addition, advanced characterization techniques such as Fourier transform infrared spectroscopy (FTIR), nuclear magnetic resonance (NMR) spectroscopy, and thermal analysis methods can provide further insights into the molecular structure, composition, and thermal behavior of the PE resin. These techniques can help in elucidating the relationships between the chemical structure of the resin and its overall properties, guiding the development of tailored formulations for specific applications. Moreover, the research also extends to the development of computational models and simulations to predict the behavior of PE resin under different processing and application conditions. This multidisciplinary approach allows for a comprehensive understanding of the resin’s behavior, enabling the design of tailored products with enhanced performance and functionality. Research on kinetic analysis and rheological methods for PE resin is a fundamental aspect of its development and optimization. By gaining insights into reaction kinetics, flow behavior, and material properties, researchers can advance the understanding of PE resin and develop tailored formulations for a wide range of applications. Rotational rheometers or rubber processing analyzers (RPAs) can be conveniently and accurately utilized to continuously measure the entire crosslinking process, including induction, curing, and over-cure periods. In the present study, the effects of PE types and peroxide complex content on crosslinking behaviors are investigated using model simulation and rheological methods. The degree of crosslinking is directly related to the stiffness of crosslinked PE. Consequently, the degree of crosslinking of PE can be conveniently evaluated using the storage modulus to establish the quantitative relationship between gel content and storage modulus. Hence, it is essential to examine the influences of different types of PE and the content of peroxide complexes on various parameters in a kinetic model, which will also be assessed to gain a deeper insight into the crosslinking behavior of PE resin.

## 2. Results and Discussion

### 2.1. Gel Content and Storage Modulus Analysis of XLPE

The S-type model refers to the curve that the composite viscosity or energy storage modulus of crosslinked polyethylene will increase with the increase of the amount of crosslinking agent or the extension of the crosslinking time. The S-type model is proposed to predict the change of crosslinking degree value according to the change of the amount of crosslinking agent or the change of crosslinking time through a mathematical empirical model. Appendix A shows the relationship between processing time and the degree of crosslinking or viscosity of the resin melt. In this study, the mathematical formula was obtained based on the crosslinking mechanism of “S” model XLPE. Therefore, the mathematical formula was called the “S” model.

Based on the crosslinking mechanism of PE resin, a straightforward and empirical mathematical “S” model was initially derived and utilized to forecast the crosslinking degree of XLPE resin by introducing peroxide mixture content. The mathematical equation is stated as: $gel\;content\;\left(\%\right)=y_{0}+C\times L\left(Z\right)$, where L(x) is known as the Langevin function, Z = X − Xc, and $y_{0}$, C and Xc are the parameters of crosslinking behaviors.

From experimental data, all parameters can be calculated well and applied to predict the degree of crosslinking for XLPE resin. In order to further investigate the feasibility of the above “S” model for different types of XLPE resins, the effects of gel content according to the peroxide mixture content are shown in Figure 1. All relationships between gel content and the peroxide mixture content for different types of XLPE resins can fit the above mathematic “S” model well. The detail of parameter values is evaluated and listed in Table 1. Therein, the XL7042 sample can provide the highest value of ideal maximum degree of crosslinking, and the XL8920 sample provides the lowest value among these four types of XLPE resins. According to the equation of gel content, the ideal maximum gel content is about 85 wt.% for HDPE 8007, which is nearly consistent with the experimental value of the XL8007 sample shown in Figure 1.

The optimum level of crosslinking for different types of XLPE resins is highly dependent on density and molecular weight. As shown in Table 1, line low-density polyethylene (LLDPE 7042) with a low melt index is particularly suited for achieving a high degree of crosslinking. Compared with HDPE 8920 and 8007, LLDPE 7042 contains more tertiary butyl chains, which are more readily attracted by free radicals. PE resins with a low melt index and longer chains can easily intertwine to form a polymeric structure, resulting in the formation of a gel structure when reacting with free radicals. For HDPE 8920, the gel content of the XL8920 sample sharply rises as the peroxide mixture content increases up to 10 wt.%, after which the curve gradually plateaus (Figure 1). This outcome is akin to the effects of silane-induced crosslinking and gamma crosslinking [20,22]. Specifically, at lower peroxide mixture contents, the crosslinking process can be rapidly promoted by free radicals generated from peroxides. However, as the concentration increases and the level of free radicals reaches equilibrium, the additional concentration contributes little to the crosslinking rate of HDPE 8920.

The conventionally used method for measuring the crosslinking degree is gel content, which typically involves the use of xylene as the solvent. However, this process is time-consuming and can also pose health risks. As a result, many researchers and engineers now prefer to use storage modulus analysis by the rheological method as an alternative to gel content. This entails establishing a mathematical equation relationship between the storage modulus (G′) value and gel content. In our research, as shown in Figure 2, a clear linear mathematical equation was established between the crosslinking degree and log (G′). Furthermore, a constant linear slope was observed between log (G′) and gel content, which exhibits some dependence on the structure of PE resins. Consequently, the plateau storage modulus (G′_max_) value can be relied upon to predict the degree of crosslinking for PE resins. Additionally, the G′_max_ value increases as the peroxide mixture content rises, as depicted in Figure 3. It is important to note that the G′_max_ value is influenced not only by crosslinking but also by the molecular weight and density of PE resin. The relationship between the G′_max_ value and peroxide mixture content closely mirrors the changes in gel content with increasing peroxide mixture content. As mentioned above, there is a linear relationship between log (G′) and gel content.

### 2.2. Crosslinking Rate of XLPE

The crosslinking rate is a measure of the reaction efficiency for XLPE resins. Moreover, an enhanced crosslinking rate can lead to a reduction in processing time. For XLPE resin, the time-dependent behavior of crosslinking can be characterized well by the following equation:(1)$$\varphi \left(t\right)=({G^{\prime }}_{\left(t\right)}-{G^{\prime }}_{0})/({G^{\prime }}_{max}-{G^{\prime }}_{0})$$

The boundary condition of this function ($\varphi \left(t\right)$) is $\varphi \left(0\right)=0$ and $\varphi \left(t\to \infty \right)=1$. As depicted in Figure 4, it exhibits an exponential growth pattern when varying the peroxide mixture content. The crosslinking behavior can be conceptualized as involving multiple chemical reactions linked to a thermodynamic function [29]. It is observed that the crosslinking rate escalates with an increasing peroxide mixture content, which can enhance the crosslinking efficiency of PE resin. However, as can be seen in Figure 4, the crosslinking rates of XLPE resins were found to show no significant disparity when different types of XLPE resins were subjected to increasing peroxide mixture content, but the influence of temperature on the crosslinking rates was observed to be significant. It was also observed that the crosslinking rates were strongly influenced by the peroxide concentration, regardless of the structure of the polyethylene (PE) resin. This suggests that the chemical composition and molecular structure of the XLPE resins do not play a significant role in determining the crosslinking rates, as the peroxide concentration had a more pronounced effect. Furthermore, this study showed that as the peroxide concentration increased, the crosslinking rates also increased, indicating a direct correlation between the two. This finding highlights the importance of peroxide concentration in controlling the crosslinking rates of XLPE resins. The study also revealed that the crosslinking rates of XLPE resins were not only influenced by the peroxide concentration and temperature, but also by the molecular weight and branching of the polyethylene resin. This suggests that the crosslinking process involves complex interactions between the peroxide, temperature, and resin properties. Moreover, the study demonstrated that the crosslinking rates of XLPE resins could be tailored by adjusting the peroxide concentration and temperature, providing a means to control the crosslinking process and customize the properties of the XLPE resins for specific applications.

### 2.3. Crosslinking Behavior Simulation of XLPE

In order to gain a better understanding of the crosslinking behavior of XLPE resin with varying peroxide mixture contents, Ding et al. [20] proposed a simplified yet realistic model. The model is represented by the following equation:(2)$$G^{\prime }\left(t\right)=A_{0}\times k2\times \frac{\left[\exp\left(-k2\times t\right)-\exp\left(-k6\times t\right)\right]}{\left[\left(1+phi\right)\left(k6-k2\right)\right]},phi=k5/k3$$

It effectively describes the relationship between crosslinking behavior and cure time (t). A_0_ indicates the concentration of the active curing agent, suggesting that higher A_0_ levels correspond to a higher degree of crosslinking. Additionally, k2 and k6 function as rate constants for the reaction from crosslink precursors to their activated forms and the reaction of crosslink degradation, respectively. The phi value is the ratio of the rate constant k5/k3, which plays a role in regulating the competition between crosslinking and side reactions by temperature. Herein, the cure time (t) is eliminated with the introduction time (ti) implying scorch time. In this mathematical model, various parameters have been assessed for XLPE resins and peroxide mixture content. Table 2 displays the fitting parameters ti, A_0_, k2, k6, and phi of different XLPE resins with the peroxide mixture content through model simulations. The values of ti, indicating the scorch time of crosslinking, increase with the rising level of the peroxide mixture content. A_0_ values, reflecting the degree of crosslinking, are highly dependent on the peroxide mixture content. k2, a function of temperature, governs the crosslinking rate. It has a higher value, indicating a faster rate of crosslinking. It is observed that k2 increases with the escalating level of the peroxide mixture content. The rate constant k6 also influences the maximum degree of crosslinking and alters the reversion rate during the over-crosslinking period. The fitting results indicate that k2 is significantly higher than k6, signifying that the precursor reaction for crosslinking activation occurs more swiftly than the crosslinking degradation process. However, k6 decreases with the increasing level of the peroxide mixture content, suggesting that the cure reversion would be less severe at higher peroxide mixture contents. The change in phi correlates to a shift in the maximum crosslinking density with crosslinking temperature. No significant differences were observed at the same temperature for different XLPE resins. A XLPE sample with high molecular weight and low density may demonstrate a low ti value, a high A_0_ value, and a low k6 value. The simulation findings indicate that high molecular weight and low-density PE resins result in a high level of crosslinking, but can also lead to premature and excessive crosslinking. However, there is no significant difference in k2 and phi values. The rate of crosslinking is largely influenced by the crosslinking temperature and peroxide mixture content, rather than the polymeric structure of PE resin.

### 2.4. Melt Processibility of XLPE by Rheological Measurement

Prior to crosslinking, PE resin must be melted to produce a dense material suitable for the rotational molding process. At this stage, the temperature will rise from room temperature to a specified temperature (200 °C), making PE resin with low activation energy more desirable for the melting process. Conversely, an increase in temperature may lead to premature crosslinking. Therefore, it is crucial to investigate the melt processibility of different types of XLPE resins with varying peroxide mixture contents through rheological measurement. To the best of our knowledge, there are few studies on the effects of PE resins and peroxide mixture content on melt processibility prior to crosslinking.

Figure 5 depicts the storage modulus versus temperature for XLPE resins with varying peroxide mixture contents. The loss modulus versus temperature for XLPE resins with varying peroxide mixture contents are displayed in Appendix A. The majority of peroxides exhibit a very brief half-life above 170 °C, and their recommended processing temperatures do not exceed 140 °C, as reported in previous studies [26,27]. Consequently, their applicability in the rotational molding process of XLPE resin is restricted. Interestingly, none of the XLPE samples examined in this study underwent premature crosslinking at temperatures below 180 °C. This finding is crucial and advantageous for the rotational molding process.

## 3. Conclusions

The crosslinking behavior of XLPE resin was examined using model simulation and the rheological method. In this study, a simple and empirical mathematical “S” model was developed to predict the degree of crosslinking of XLPE resin. According to this model, the optimal maximum gel content for HDPE is approximately 85%. XLPE resin with high molecular weight and low density tends to exhibit a high degree of crosslinking, but this can also result in premature and excessive crosslinking. Except for the effect of temperature, the rate of crosslinking is largely dependent on the concentration of the peroxide, rather than the polymeric structure. The peroxide mixture content has minimal impact on the variability of the storage modulus with increasing temperature, prior to crosslinking. The melt viscosities before crosslinking for different XLPE resins are primarily determined by their melt index. Higher melt index values are associated with lower storage modulus improved melt processability during processing. XLPE resins with high molecular weight and low density exhibit a low value of ti, high value of A_0_, and low value of k6. However, there is no significant difference in the values of k2 and phi. All model simulation results showed good agreement with experimental rheological data, which are critically important for the industrial production of XLPE resins.

## 4. Materials and Methods

### 4.1. Materials

HDPE (8007), with a melt index of 7.5 g/10 min at 190 °C and density of 0.963 g/cm^3^, and LLDPE (7042), with a melt index of 2.0 g/10 min at 190 °C and density of 0.918 g/cm^3^ were both provided by Shenhua Group Ltd. from Beijing, China. MDPE (2388), with a melt index of 0.55 g/10 min at 190 °C and density of 0.941 g/cm^3^, and HDPE (8920), with a melt index of 20 g/10 min at 190 °C and density of 0.965 g/cm^3^, were separately obtained from Dow Chemical (Midland, MI, USA) and ExxonMobil (Spring, TX, USA). A peroxide mixture containing a small amount of PE resin, a particular amount of peroxide, crosslinking coagents, and processing aids was manufactured in our laboratory. All other chemicals were purchased from Sigma-Aldrich from Burlington, MA, USA. The compositions of the as-prepared different polymer materials with various amounts of the peroxide mixture are presented in Table 3. The polymer material and the peroxide mixture were pre-mixed for approximately 3 min and then compounded using an AK-36 twin-screw extruder with a diameter of 36 mm and an LD ratio of 48:1, running at a speed of 230 rpm, at 160 °C. High-pressure molding was carried out in a hot press (P300P series, Collin Inc., Austin, TX 78750, USA) at 160 °C and 12 MPa for 5 min. The sheet samples were then cooled to room temperature at a rate of 10 min.

### 4.2. Methods

The gel content, which measures the crosslinking degree of XLPE, was determined gravimetrically using xylene as the solvent at 170 °C, according to ASTM-D2765 [32]. An XLPE sample weighing 0.400 ± 0.015 g was cut into small pieces and placed in a 100 stainless steel wire cloth. The specimens were then extracted for 20 h, followed by drying in a vacuum oven at 90 °C for 6 h. The gel content of the XLPE sample was evaluated using the equation:
Gel Content (%) = (1 − W/W_0_) × 100%(3)
where W_0_ and W are the weight of the samples before and after solvent extraction, respectively. A controlled strain rheometer (TA rheometer, Lukens Drive New Castle, DE, USA) in oscillatory mode with parallel plate fixture (20 mm in diameter) at a gap of 1.0 mm was used to measure the crosslinking behavior. The processing conditions consisted of two stages: (Step 1) the temperature is increased from 140 °C to 200 °C at a rate of 10 °C/min; (Step 2) a time sweep was performed at 200 °C for 15 min. All rheological testing conditions were maintained at a 5% strain and a frequency of 1.0 Hz. The storage modulus (G′) obtained from rheological measurement for XLPE serves as an indicator of melt strength for the melting resin. During the temperature increase stage, the changes in storage modulus with increasing temperature can be used to characterize the melt processibility before crosslinking. During the time sweeping stage, the variation in storage modulus reflects the degree of crosslinking for XLPE. From this stage, all relevant parameters of crosslinking behavior, such as scorch time, crosslinking rate, and final degree of crosslinking can be accurately determined.

## Figures and Tables

**Figure 1 gels-10-00035-f001:**
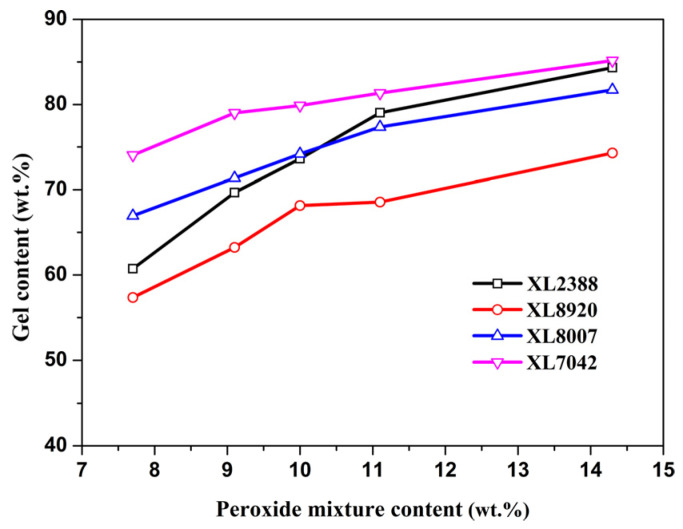
Effects of gel content according to peroxide mixture content for different types of XLPE resins.

**Figure 2 gels-10-00035-f002:**
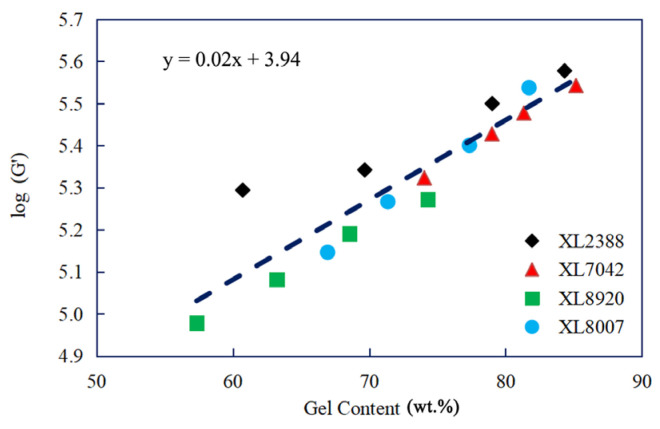
Linear relationship between log (G′) and gel content for different types of XLPE resins.

**Figure 3 gels-10-00035-f003:**
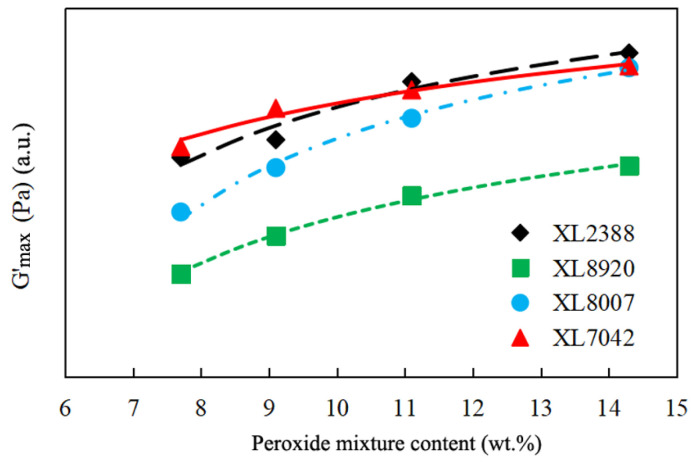
Effects of the plateau storage modulus (G′_max_) according to peroxide mixture content for different types of XLPE resins.

**Figure 4 gels-10-00035-f004:**
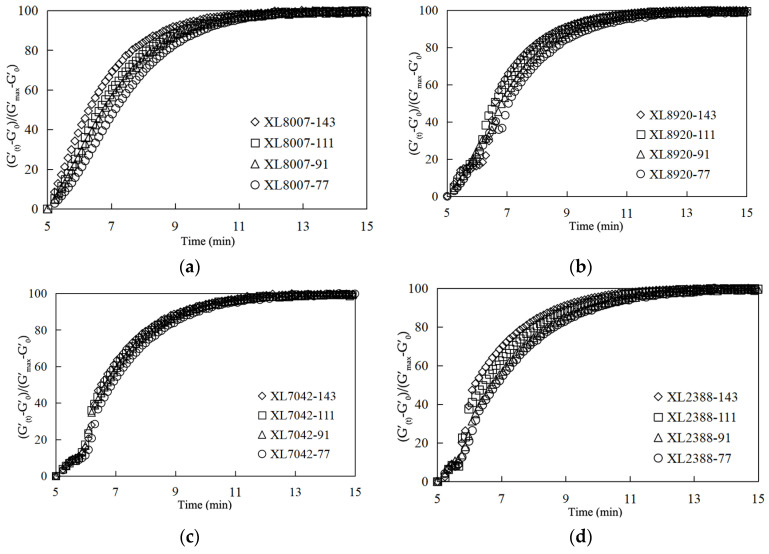
Effects of crosslinking rate according to peroxide mixture content for different types of XLPE resins: (**a**) HDPE 8007; (**b**) HDPE 8920; (**c**) LLDPE 7042; (**d**) MDPE 2388.

**Figure 5 gels-10-00035-f005:**
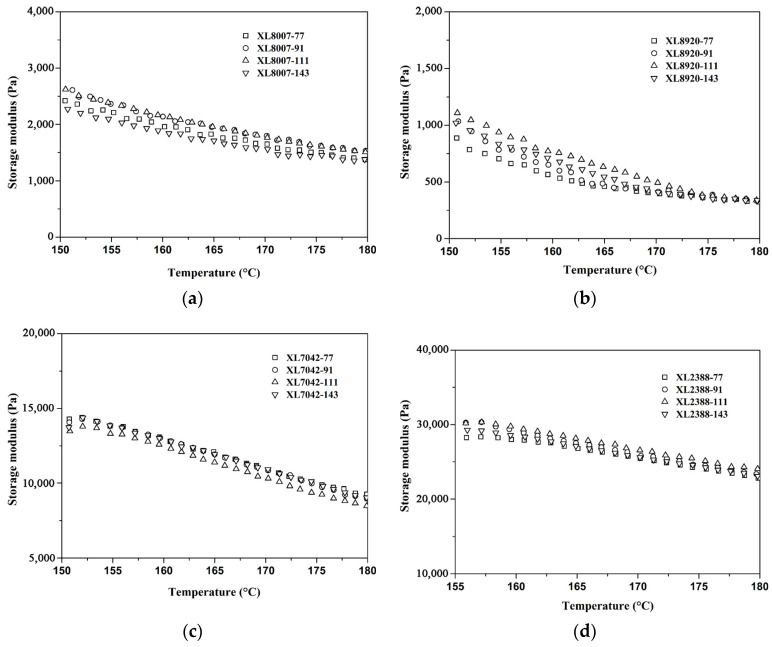
Effects of storage modulus according to peroxide mixture content for different types of XLPE resins: (**a**) HDPE 8007; (**b**) HDPE 8920; (**c**) LLDPE 7042; (**d**) MDPE 2388.

**Table 1 gels-10-00035-t001:** Parameter values for different types of XLPE.

Parameter	XL2388	XL8920	XL8007	XL7042
$y_{0}$	40.53	35.33	38.24	39.22
$x_{c}$	6.38	5.9	5.46	4.51
C	48.11	42.58	46.9	50.38
Ideal maximum degree of crosslinking	89	78	85	90
Adj. q-square	0.997	0.997	0.997	0.999

**Table 2 gels-10-00035-t002:** The fitting results of ti, $A_{0}$, k2, k6, and phi of different types of polyethylene resins.

Parameter	Peroxide Mixture Content (wt.%)	XL7042	XL8007	XL8920	XL2388
ti (s)	7.7	0.22	0.25	0.31	0.22
	9.1	0.29	0.32	0.30	0.12
	11.1	0.31	0.43	0.30	0.10
	14.3	0.41	0.51	0.32	0.16
A_0_ (Pa)	7.7	238,972	158,665	107,356	217,904
	9.1	279,737	199,809	133,582	235,183
	11.1	309,557	263,437	160,588	322,242
	14.3	346,514	342,194	194,874	380,893
k2	7.7	0.44	0.37	0.38	0.37
	9.1	0.52	0.44	0.44	0.42
	11.1	0.55	0.49	0.56	0.57
	14.3	0.65	0.66	0.56	0.68
k6	7.7	0.0086	0.0111	0.0108	0.0089
	9.1	0.0051	0.0075	0.0104	0.0059
	11.1	0.0027	0.0037	0.0024	0.0019
	14.3	0.0015	0.0008	0.0031	0.0011
phi	7.7	6.1403 × 10^−5^	6.1403 × 10^−5^	6.1403 × 10^−5^	6.1403 × 10^−5^
	9.1	6.1403 × 10^−5^	6.1403 × 10^−5^	6.1403 × 10^−5^	6.1403 × 10^−5^
	11.1	6.1403 × 10^−5^	6.1403 × 10^−5^	6.1403 × 10^−5^	6.1403 × 10^−5^
	14.3	6.1403 × 10^−5^	6.1403 × 10^−5^	6.1403 × 10^−5^	6.1403 × 10^−5^

**Table 3 gels-10-00035-t003:** The compositions of different polymer materials.

PE Resin Type	PE Resin Code	PE Resin Content (wt.%)	Peroxide Mixture Content (wt.%)
HDPE 8007	XL8007-77	92.3	7.7
	XL8007-91	90.9	9.1
	XL8007-111	88.9	11.1
	XL8007-143	85.7	14.3
LLDPE 7042	XL7042-77	92.3	7.7
	XL7042-91	90.9	9.1
	XL7042-111	88.9	11.1
	XL7042-143	85.7	14.3
HDPE 8920	XL8920-77	92.3	7.7
	XL8920-91	90.9	9.1
	XL8920-111	88.9	11.1
	XL8920-143	85.7	14.3
MDPE 2388	XL2388-77	92.3	7.7
	XL2388-91	90.9	9.1
	XL2388-111	88.9	11.1
	XL2388-143	85.7	14.3

## Data Availability

Data are contained within the article and Appendix A.

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
