# Peer review of "Model Simulation and Rheological Research on Crosslinking Behavior of Polyethylene Resin"

_gels, 2023, doi:10.3390/gels10010035_

Round 1

Reviewer 1 Report

Comments and Suggestions for Authors

This manuscript deals with the crosslinking of various polyethylene samples by a peroxide mixture. It is recommended to accept this paper for publication after some revision on the basis of comments below.

COMMENTS

1. Why the authors call the mathematical formula as “S” model?

2. In lines 82-85, the authors write the following: “Based on the crosslinking mechanism of PE resin, a straightforward and empirical mathematical "S" model is initially derived and utilized to forecast the crosslinking degree 83 of XLPE resin by introducing peroxide mixture content. The mathematical equation is 84 stated as follows”.

Where this “straightforward and empirical mathematical "S" model” comes from, and on what basis?

What “initially derived” means here? There is no any explanation, background or model calculation is presented here for equations (1) and (2).

3. In line 86, the authors write that “y0, C and Xc are the parameters of crosslinking behaviors”. What is the physical meaning of these parameters?

4. In line 92, the authors write that “The detail of parameter values is evaluated and listed in Table 1.” Where these parameter values are come from, how were these evaluated?

5. In line 96, correctly HDPE and not  HPDE.

6. In lines 95-97, the authors claim that “According to Equation (1), the ideal maximum gel content is about 85 wt.% for HPDE 8007, which is nearly consistent with the experiment value of the XL8007 sample in Figure 1.”

Where this conclusion comes from? The highest gel content value of XL8007 in Figure 1 is significantly lower than 85 wt.%.

7. How come that the y0, C and Xc parameters are significantly different between the polyethylene samples? What is the explanation of the authors for the big differences between these, presumably universal parameters for all the HDPE materials?

8. The rheological curves, including both G’ and G’’, for all the samples which were used to determine the storage modulus should be presented in this manuscript in either the main text or as Supporting Information.

9. The units for the data in Table 2 must be provided.

10. In lines 232-233, the authors write that “The peroxide mixture containing a small amount of PE resin, a certain amount of peroxide, crosslinking coagents, and processing aids is manufactured in our laboratory”. The authors should precisely describe what kind of peroxide, crosslinking coagents and processing aids, and in what ratio were used for obtaining the applied “peroxide mixture”, and how exactly were these prepared.

Author Response

Pls see the attached PDF. Thanks a lot for your suggests.

Xuelian CHEN 

Reviewer 2 Report

Comments and Suggestions for Authors

Comments on the Quality of English Language

The English is acceptable. The grammar and syntax are largely correct, though there are occasional opportunities for simplifying sentence structures to enhance readability. Technical terminology should be improved. The consistent academic style and tone are suitable for a scientific research paper, and overall, the paper maintains fair readability. Moderate revisions could further improve clarity, particularly in dense technical sections, ensuring a balance between technical accuracy and accessibility for a broader audience.

Author Response

Please see the attached document. thanks for your suggestions. 

Xuelian CHEN 

Round 2

Reviewer 2 Report

Comments and Suggestions for Authors

The authors have addressed my questions. The manuscript has been improved. The only suggestion I have is for equations 1 and 2. Equation 1 can be written as: gel content (%) = y0 + C*L(Z), where L(x) is known as the Langevin function, and Z=X-Xc